# Neuroprotective Effects of Chlorogenic Acid in a Mouse Model of Intracerebral Hemorrhage Associated with Reduced Extracellular Matrix Metalloproteinase Inducer

**DOI:** 10.3390/biom12081020

**Published:** 2022-07-22

**Authors:** Yang Liu, Fei Wang, Zhe Li, Yanling Mu, Voon Wee Yong, Mengzhou Xue

**Affiliations:** 1Department of Cerebrovascular Diseases, The Second Affiliated Hospital of Zhengzhou University, Zhengzhou 450001, China; liuy@zzu.edu.cn (Y.L.); wangfei3871@126.com (F.W.); librocoli@163.com (Z.L.); muyanling11@163.com (Y.M.); 2Henan Medical Key Laboratory of Translational Cerebrovascular Diseases, Zhengzhou 450001, China; 3Department of Clinical Neurosciences, Hotchkiss Brain Institute, University of Calgary, Calgary, AB T2N 1N4, Canada

**Keywords:** intracerebral hemorrhage, chlorogenic acid, extracellular matrix metalloproteinase inducer, matrix metalloproteinase-9

## Abstract

Chlorogenic acid (CGA) has been reported to have various biological activities, such as anti-inflammatory, anti-oxidant and anti-apoptosis effects. However, the role of CGA in intracerebral hemorrhage (ICH) and the underlying mechanisms remain undiscovered. The current study aims to investigate the effect of CGA on neuroinflammation and neuronal apoptosis after inhibition of EMMPRIN in a collagenase-induced ICH mouse model. Dose optimization data showed that intraperitoneal administration of CGA (30 mg/kg) significantly attenuated neurological impairments and reduced brain water content at 24 h and 72 h compared with ICH mice given vehicle. Western blot and immunofluorescence analyses revealed that CGA remarkably decreased the expression of extracellular matrix metalloproteinase inducer (EMMPRIN) in perihematomal areas at 72 h after ICH. CGA also reduced the expression of matrix metalloproteinases-2/9 (MMP-2/9) at 72 h after ICH. CGA diminished Evans blue dye extravasation and reduced the loss of zonula occludens-1 (ZO-1) and occludin. CGA-treated mice had fewer activated Iba-1-positive microglia and MPO-positive neutrophils. Finally, CGA suppressed cell death around the hematoma and reduced overall brain injury. These outcomes highlight that CGA treatment confers neuroprotection in ICH likely by inhibiting expression of EMMPRIN and MMP-2/9, and alleviating neuroinflammation, blood–brain barrier (BBB) disruption, cell death and brain injury.

## 1. Introduction

Intracerebral hemorrhage (ICH) is a catastrophic cerebrovascular disease accounting for approximately 12 to 20% of strokes [1,2]. It has up to 50% mortality and 70% disability among survivors 1 year after the onset of the disease [1]. Despite the advances of minimally invasive surgery for primary ICH injury that has reduced neurological deficits and mortality, the prognosis remains poor due to the combined effects of primary and secondary injuries following ICH [3,4,5]. Clinical studies on neuroprotective agents have largely failed [6], and there are currently no available medical treatments that sufficiently benefit patients’ prognosis and neurological functions [7,8,9]. Mounting evidence demonstrates that excessive inflammation drives secondary brain injury of ICH, including neuronal death, increased blood– brain barrier (BBB) permeability and the release of inflammatory cytokines [5,10,11,12]. Therefore, modulating the inflammatory microenvironment within the brain may be a promising therapeutic strategy after ICH.

Extracellular matrix metalloproteinase inducer (EMMPRIN, CD147), a trans-membrane glycoprotein of the immunoglobulin (Ig) superfamily, can trigger the production of matrix metalloproteinases (MMPs); it has a key role as a mediator of inflammation and immune responses [13]. Following brain injury, many cells such as neurons, monocytes/macrophages, astrocytes and neutrophils can secrete MMP-9 [14,15,16]. MMP-9 can degrade extracellular matrix proteins and exacerbates inflammatory responses [17]. Notably, following middle cerebral artery occlusion (MCAO) in animal models, the blockade of EMMPRIN reduced MMP-9 expression, diminished brain injury and improved long-term cognitive outcomes [18]. We reported that EMMPRIN and MMP-9 are upregulated after ICH, and that minocycline has neuroprotective effects associated with the inhibition of expression of EMMPRIN and MMP-9 [19,20,21].

Chlorogenic acid (CGA, 3-O-caffeoylquinic acid) is a dietary phenylpropanoid molecule; it is widely distributed in many kinds of natural products such as tea, coffee, fruits and vegetables [22]. Importantly, CGA can cross the BBB and is beneficial in the treatment of certain neurological disorders [23]. Numerous studies have demonstrated that CGA has many beneficial effects including anti-oxidative and anti-inflammatory outcomes [24,25,26,27]. Previously, in an ischemia/reperfusion (I/R) injury rat model, CGA confers neuroprotection of global ischemic insult by suppressing the expression of multiple inflammatory factors; this is accompanied by reduced levels of glutamate, calcium and nitrate in different brain regions and CSF [28]. Recent evidence indicates that CGA inhibits MMP-9/2 expression and activity, and reduces brain injury in a rat model of transient middle cerebral artery occlusion (tMCAO) [29]. While these studies corroborate the neuroprotective ability of CGA, its precise effects and mechanisms of potential neuroprotection in ICH are not explored.

In the present study, we have addressed the effects of CGA in a mouse model of ICH. Moreover, we investigated whether CGA could reduce the expression of EMMPRIN, neuroinflammation, and cell death.

## 2. Materials and Methods

### 2.1. Animals

Adult male C57BL/6 mice (20–23 g) were provided by Beijing Vital River Experimental Animals Centre (Beijing, China). The experimental protocols and animal experiments were approved by the Ethics Committee of Zhengzhou University, according to the China Council on Animal Care guidelines. All animals had been housed under controlled conditions in a Specified Pathogen Free (SPF) room (12 h light/dark cycle, temperature 25 °C, relative humidity (RH) 55%), and free access to food and water.

### 2.2. Intracerebral Hemorrhage Model of Mice

ICH mouse model was prepared based on the previous literature [30]. Mice were anesthetized with pentobarbital (40 mg/kg) by intraperitoneally (ip) injection. Thereafter, mice were positioned on a stereotactic frame (RWD, Shenzhen, China). An incision was made in the scalp and the skull was exposed extensively. A small hole was drilled on the skull’s right side of (coordinates: 2.0 mm lateral to midline, 0.8 mm anterior to the bregma). Bacterial collagenase type VII (0.075 units, Sigma-Aldrich, Milwaukee, WI, USA) was dissolved in 0.5 μL of sterile saline and infused into brain at a rate of 0.1 μL/min with an infusion pump (3.5 mm deep relative to bregma). After injection, the needle was left in the same position for 10 min to prevent reflux and then slowly withdrawn. The cranial burr hole was covered with bone wax, and the wounded scalp was sutured. Sham group were injected with an equal volume of sterile saline.

### 2.3. Experimental Design

In this study, all mice were randomly assigned to one of five experimental procedures (Figure 1). Animals that died during the experiment were not included in the experimental study group.

In this study, we chose the chlorogenic acid (CGA) dose based on the previous study [29]. Mice received CGA (Sigma-Aldrich, USA, purity > 95%) or phosphate-buffered saline (PBS) intraperitoneally 1 h after ICH and once a day until euthanasia. The CGA was dissolved in phosphate-buffered saline (PBS) solution (every 3 mg CGA was dissolved in 1 mL PBS).

#### 2.3.1. Experiment 1

A dose optimization study was performed to determine the effective dose of CGA for neuroprotection. Three doses of CGA (10 mg/kg, 30 mg/kg, and 60 mg/kg, ip) were evaluated. Mice were equally and randomly divided into five groups: a sham group, ICH + Vehicle group, ICH + CGA group (10 mg/kg, ip), ICH + CGA group (30 mg/kg, ip), and ICH + CGA group (60 mg/kg, ip). Based on brain water content and neurobehavioral tests, CGA treatment at a dose of 30 mg/kg was determined to have the best neuroprotective effect. Thereafter, to observe the effect of drug at 72 h after ICH, 18 mice were randomly divided into 3 groups: sham group, ICH + Vehicle, ICH + CGA group (30 mg/kg, ip).

#### 2.3.2. Experiment 2

To evaluate the effects of CGA on EMMPRIN and MMP-2/9 at 72 h after ICH, mice were randomly divided into three groups: sham group, ICH + Vehicle group, ICH + CGA group (30 mg/kg, ip) (*n* = 4/group).

#### 2.3.3. Experiment 3

To investigate the effects of CGA on BBB at 72 h after ICH, mice were randomly divided into sham group, ICH + Vehicle group or ICH + CGA group (30 mg/kg, ip). EB staining and Western blot was then investigated (*n* = 4/group).

#### 2.3.4. Experiment 4

To assess the effects of CGA on microglia/macrophage activation and neutrophil infiltration at 72 h after ICH, mice were randomly divided into three groups: sham group, ICH + Vehicle group, ICH + CGA group (30 mg/kg, ip) (*n* = 4/group).

#### 2.3.5. Experiment 5

To evaluate the effects of CGA on cell death and hematoma volume at 72 h after ICH, mice were randomly divided into sham group, ICH + Vehicle group, or ICH + CGA group (30 mg/kg, ip). TUNEL staining (*n* = 4/group), hematoma volume (*n* = 6/group) and Hematoxylin and Eosin staining (*n* = 3/group) were then examined.

### 2.4. Neurological Function Tests

Neurological function tests were conducted at day 1 and 3 after ICH using the Modified neurological severity score (mNSS) and corner turn test to assess motor, sensory, reflex and balance functions together, as previously described [31,32]. The mNSS test was graded from 0 (normal performance) to 18 (maximum defect). Higher scores represent more severe neurological dysfunction. The corner turn test to evaluate sensory motor and postural asymmetries. In this process, each mouse is allowed to proceed into a corner with an angle of 30° and then given the choice of turning left or right, and the direction choice was recorded. Each mouse repeated this procedure 10 times, and the percentage of left turns was calculated. All behavioral tests were performed in duplicate.

### 2.5. Analysis of Brain Cell Death

We used TUNEL staining to evaluate cell death (including necrosis and apoptosis) 3 days following ICH, according to the manufacturer’s protocol (Vazyme Biotech, Nanjing, China). Mouse brains were collected and processed as described previously for tissue preparation. For quantification of TUNEL-positive cells, four photographs with a field of view of 200× were randomly taken around the hematoma from each brain slice. These analyses were conducted blind by a researcher who did not know the experimental conditions. The ratio of dead cells to total nuclei was calculated as dead cells (%) [33].

### 2.6. Measurement of Brain Water Content

We used the wet/dry method to evaluate brain water content. Briefly, mice were deeply anesthetized at 24 and 72 h after ICH, and the brains were removed and divided into three parts quickly: ipsilateral hemisphere, contralateral hemisphere (control) and the cerebellum. Each part was weighed on an electronic balance to obtain the wet weight. The brains were dried in an oven at 100 °C for 24 h to obtain the dry weight. The brain water content was determined by the following formula: [(Wet Weight − Dry Weight)/Wet Weight] × 100%.

### 2.7. Assessment of BBB Permeability

We measured and compared the BBB permeability of each group 3 days after ICH. The mice were given 2% Evans blue (EB) solution (4 mL/kg, Sigma-Aldrich, St Louis, MO, USA) via the tail vein 3 h before the mice were sacrificed. The brains were quickly removed and divided into the left and right hemispheres, and each hemisphere was then weighed as previously described [19]. Each hemisphere was homogenized into a test tube with 5 mL of formamide (Sigma-Aldrich, St Louis, MO, USA), and followed by incubation in an oven at 60 °C for 48 h. After centrifugation at 1000× rpm for 10 min, supernatants were collected. The supernatants were analyzed at 630 nm with an absorbance reader (Molecular Devices, San Jose, CA, USA) to record optical density (OD). The standard curve is obtained by measuring the standard product (50, 25, 12.5, 6.25, 3.125, 1.5625, 0.78125, 0.39062, 0.19531, 0.09766 μg/mL), then concentration of EB is calculated according to the standard curve.

### 2.8. Immunofluorescence

Immunofluorescence staining methods were described previously [19,21]. The samples were incubated with rabbit anti-EMMPRIN monoclonal antibody (1:200, Abcam, Cambridge, MA, USA), MPO polyclonal antibody (1:300, Abcam, Cambridge, MA, USA) and Iba-1 polyclonal antibody (1:300, Wako, Osaka, Japan). Fluorescent images were obtained by using an OLYMPUS microscope (BX53-P, Olympus Co., Tokyo, Japan). For positive cell counting, four microscopic fields in each brain section and three sections per mouse were examined.

### 2.9. Western Blot Analysis

Ipsilateral hemispheres protein homogenate was extracted from mouse brain 3 days after ICH. Western blotting was performed as we previously described [21]. Proteins were electrophoresed and transferred onto a PVDF membrane (Merck KGaA, Darmstadt, Germany). After being blocked, membranes were incubated overnight at 4 °C with mouse anti-EMMPRIN monoclonal antibody (1:1000, Santa Cruz Biotechnology, Santa Cruz, CA, USA), rabbit anti-MMP-9 polyclonal antibody (1:1000, Abcam, Cambridge, MA, USA), rabbit anti-ZO-1 polyclonal antibody (1:1000, Proteintech, Wuhan, China), rabbit anti-occludin polyclonal antibody (1:1000, Abcam, Cambridge, MA, USA), and GADPH antibody (1:5000, Proteintech, Wuhan, China). Afterward, membranes were incubated for 1 h at room temperature with the HRP-conjugated anti-rabbit IgG antibody (Servicebio, Wuhan, China) or the anti-mouse IgG antibody (Servicebio, Wuhan, China). Finally, the protein-specific signals were detected using a BIO-RAD ChemiDoc™ MP Imaging System.

### 2.10. Hematoma Volume Measurement

Hematoma volumes were quantified at day 3 after ICH, as we previously described [21]. The fixed brain specimens were coronally sectioned with a thickness of 1 mm. The volume of hematoma was calculated using the formula V = t × (A1 + A2 + … + An), where V, t and A represented the hematoma volume, the thickness of the section and the bleeding area, respectively.

### 2.11. Hematoxylin and Eosin (H&E) Staining

HE staining was used to assess brain damage. The brain tissues were cut into 5 μm thick paraffin-embedded sections, after that the sections were dried at 50 °C for at least 30 min. Then the sections were treated as we did in a previous study [19].

### 2.12. Statistical Analysis

All data were presented as mean ± SD. Blinding investigation was applied in all analyses. One-way analysis of variance (ANOVA) followed by Tukey’s multiple comparisons test was used to compare differences between multiple groups. A difference of *p* < 0.05 indicated statistical significance. GraphPad 6.0 (GraphPad Software Inc., San Diego, CA, USA) was utilized in all statistical analyses.

## 3. Results

### 3.1. CGA Treatment Attenuated Neurological Impairments and Brain Edema at 24 h and 72 h after ICH

In order to choose the optimal dose to attenuate ICH-induced brain injury, three different doses of CGA were tested. Our results showed that the brain water content in the ipsilateral hemisphere was significantly increased in the ICH + vehicle groups when compared with sham group at 24 h after ICH (Figure 2A), which was significantly reduced after CGA treatment (30 mg/kg) (Figure 2A).

We analyzed behavioral deficits and these were more severe in ICH + vehicle group than in sham group at 24 h as assessed by the mNSS (Figure 2B) and corner turn test (Figure 2C). Treatment with CGA (30 mg/kg) significantly improved neurological impairments when compared with ICH + vehicle group at 24 h after ICH (Figure 2B,C). To confirm the efficacy of CGA (30 mg/kg), we examined brain edema and behavior tests at 72 h after ICH. ICH + CGA (30 mg/kg) group had significantly improved neurological impairments and reduced brain water content in the ipsilateral hemisphere when compared to the ICH + vehicle group (Figure 2D–F). Thus, we chose a dosage of CGA at 30 mg/kg for subsequent studies.

### 3.2. CGA Treatment Reduced the Expression of EMMPRIN and MMP-2/9

To assess whether CGA affected EMMPRIN expression following ICH, we measured changes in EMMPRIN expression in the perihematomal area. The number of EMMPRIN-positive cells and protein expression level of EMMPRIN in the perihematomal tissue at 72 h after ICH were significantly upregulated in the ICH + vehicle group compared with the sham group, and this was significantly reduced after CGA treatment (Figure 3A–D). Additionally, the expression of MMP-2/9 in the perihematomal area at 72 h were measured by Western blot. The results show that MMP-2/9 were significantly increased in ICH + vehicle group compared to sham injury, and this was significantly decreased with CGA treatment (Figure 3E–G).

### 3.3. CGA Reduced BBB Permeability after ICH

EB staining demonstrated that CGA treatment diminished Evans blue dye extravasation compared with the ICH + vehicle (Figure 4A,B), at 72 h after ICH. To further elucidate the observed improvements in the BBB, we used Western blots to measure the expression of tight junction proteins (ZO-1 and occludin). The results show decreased expression of ZO-1 and occludin resulting from ICH, and that mice that received CGA after ICH had higher expression of ZO-1 and occludin than those in the ICH + vehicle group (Figure 4C–F).

### 3.4. CGA Treatment Alleviated Microglia/Macrophage Activation and Neutrophil Infiltration after ICH

Inflammation contributes to ICH brain injury. We examined microglia/macrophage activation and neutrophil infiltration by immunofluorescence staining at 3 days after ICH. Our results indicate that the numbers of Iba-1 and MPO-positive cells in the perihematomal area were significantly increased in ICH + vehicle group when compared with sham group at 3 days after ICH (Figure 5A–D). However, in the group treated with CGA, the number of Iba-1 and MPO-positive cells dropped below those of the ICH + vehicle group (Figure 5A–D). These results demonstrate that CGA attenuated inflammation after ICH.

### 3.5. CGA Treatment Reduced Cell Death and Brain Injury after ICH

TUNEL staining was used to assess cell death 3 days after ICH. While the number of dead cells increased in the perihematomal area in ICH + vehicle group compared to sham, CGA treatment significantly reduced cell death (Figure 6A,B). Furthermore, mice treated with CGA presented with smaller lesion volume compared with the ICH + vehicle animals (Figure 6C,D). H&E staining further indicated that after injection of saline, there was a small amount of blood and negligible edema around the striatum (Figure 6E). In the ICH group, there were a large number of red blood cells, loose brain tissue around the hematoma and more leukocyte infiltration. After CGA treatment, the above symptoms were significantly improved (Figure 6E). These results show that CGA can effectively reduce brain injury after ICH.

## 4. Discussion

ICH is a devastating injury that brings a severe burden on society and families. Most patients with ICH will retain varying degrees of disability for the rest of their life. Despite this dismal prognosis, the drugs currently available to treat ICH are very limited and have poor efficacy [34]. Evidence suggests CGA is a neuroprotective agent [35,36], and has neurotrophic [27], anti-inflammatory [37] and antioxidant properties [38]. Therefore, we explored the efficacy of CGA treatment after ICH in a mouse model. Our results suggest that CGA has neuroprotective effect on ICH. First, CGA promoted neurological recovery and reduced brain water content at different doses, with best dose at 30 mg/kg. Second, treatment with CGA was associated with downregulation of EMMPRIN and MMP-2/9, and reduced blood–brain barrier disruption. Third, CGA inhibited microglia/macrophage activation and neutrophil infiltration, suppressed brain cell death in perihematomal areas and minimized lesion volume after ICH. These findings reveal that CGA can protect against brain injury, and EMMPRIN may be a target for therapy after ICH.

Studies demonstrate the main causes of ICH progression are neuroinflammation and oxidative stress [11]. However, some studies have indicated that neuroinflammation promote repair by providing neurotrophic factors and removing inhibitors in the lesion that otherwise impede recovery. However, considering that the beneficial aspects of neuroinflammation that manifest more prominently in the later periods after ICH, the treatment of secondary injury caused by inflammatory response should be limited to the early stage of injury [11]. In this study, we focused mainly on the treatment of chlorogenic acid in the acute phase after ICH. Numerous studies have showed that EMMPRIN is pro-inflammatory, and that it can induce MMPs expression in neurological diseases leading to aggravation of neurological injury [18,39,40]. Our previous work showed that both the number of positive cells and protein expression of EMMPRIN were significantly increased, along with induced expression of MMP-9 after ICH in mice [19,21]. Increasing evidence has shown its strong neuroprotective effect by blocking EMMPRIN upregulation to reduce neuroinflammation in experimental cerebral ischemia [18,40]. One study reported that inhibition of EMMPRIN expression significantly decreased pro-inflammatory microglia/macrophage markers, but did not alter regulatory microglia/macrophage markers in tMCAO-induced injury [39]. However, the potential therapeutic effects of targeting EMMPRIN in ICH remain to be elucidated.

CGA is a common compound in tea, coffee, fruits and vegetable [22]. It has anti-inflammatory, MMP-2/9 inhibitory [29] and anti-oxidative outcomes [38]. The bioavailability of oral chlorogenic acid is poor due to first pass metabolism by the liver and degradation by normal bacteria in the digestive tract [41]. The highest plasma concentration was attained 13.33 min after intravenous treatment, and its half-life was 59.1 min after intraperitoneal administration [42,43]. Previous studies have shown that intraperitoneal injection of 30 mg/kg CGA improves focal ischemia-induced brain injury in tMCAO model [29]. In this study, pure CGA compound was administered intraperitoneally 1 h after ICH. After an initial dose finding, we found that CGA at 30 mg/kg has the best neuroprotective effect. This is consistent with previous research findings [29]. Accordingly, we found that CGA treatment attenuated neurological dysfunction and reduced brain edema at 24 h and 72 h after ICH.

Previous studies have reported the involvement of CGA in the regulation of inflammatory processes [29]. Enhanced MMP activity, especially MMP-2/9, plays a key role in the pathophysiology of secondary brain injury following ICH [44,45]. MMP-9 is not only involved in the breakdown of the BBB, but also in edema formation, hematoma enlargement, as well loss of neurons after hematoma [15]. Therefore, MMP-9 inhibition is thought to be a potential target for ICH therapy. Several different therapeutic approaches have been tried, but no MMP-9 blocking treatment has been developed due to the lack of specific MMP-9 inhibitors [46]. Previous studies have found that CGA inhibited MMP-2/9 activity [29,47]. EMMPRIN is an inducer upstream of MMP-9 and inhibition of EMMPRIN may be an effective alternative to inhibit MMPs activity after ICH. In this study, we found that both number of positive cells and protein expression of EMMPRIN were significantly reduced after CGA treatment, while MMP-2 and MMP-9 protein expression were downregulated; this suggests CGA as a new therapeutic approach to affect MMPs.

In our previous study, we found that endothelial cells, astrocytes and microglia/macrophages in the brain express higher levels of EMMPRIN after ICH. In addition, EMMPRIN could mediate the induction of MMP-9 expressions in these brain cell types after ICH [21]. MMP-2/9 are involved in BBB disruption. Previous studies have found that CGA protects the integrity of the BBB by inhibiting the expression of MMP-2 and MMP-9 mRNA and protein [29]. In this study, we confirm these results. Moreover, we found that CGA treatment decreased Evans blue extravasation and reduced brain water content suggesting the protection of the BBB. MMP-2/9 are causes of cell death after ICH since direct use of MMP-2/9 can lead to neuronal death in cell culture studies [44,48]. The phenomenon of programmed cell death, known as anoikis, occurs in cell-dependent cells when extracellular matrix degradation interferes with cell-matrix interactions, cell attachment and integrin signaling [49,50,51]. Based on these reports, we can speculate that MMPs are harmful factors directly contributing to neuronal injury in addition to brain edema. Therefore, our findings suggest that the effect of CGA on MMP-2 and -9 expression may be the mechanism for its protective effect against neuronal injury and BBB disruption.

Microglia/macrophage activation and neutrophil infiltration following ICH can further promote the release of pro-inflammatory mediators and molecular markers of neuronal apoptosis, ultimately leading to neuroinflammation and neuronal apoptosis [12]. Immunofluorescence staining showed a significant reduction in microglia activation and neutrophil infiltration following CGA treatment of ICH. Our results thus suggest that the neuroprotective effects of CGA could also be contributed by its ability to reduce neuroinflammation.

There are some limitations to our study. Firstly, our study focused mainly on the effect of chlorogenic acid in the acute stage of ICH, so its long-term efficacy after ICH is unknown. Secondly, the BBB permeability increased after ICH, detailed roles of EMMPRIN on other infiltrating cells, such as neutrophils, are necessary for further study. Thirdly, only adult male mice were examined, so the results would need to be tested in female subjects. Last but not least, due to the use of bacterial collagenase to induce ICH in this study, we cannot rule out the possibility of triggering greater neuroinflammation than in the ICH model induced by autologous blood. Therefore, further studies are needed to investigate the neuroprotective effects of using these two models to block EMMPRIN with chlorogenic acid after ICH.

## 5. Conclusions

In conclusion, we demonstrate that chlorogenic acid exerts neuroprotective roles after ICH. It inhibited the expression of EMMPRIN and MMP-2/9, it alleviated neuroinflammation and BBB damage and it promoted neurological function recovery. These findings hold promise for the treatment of patients with ICH.

## Figures and Tables

**Figure 1 biomolecules-12-01020-f001:**
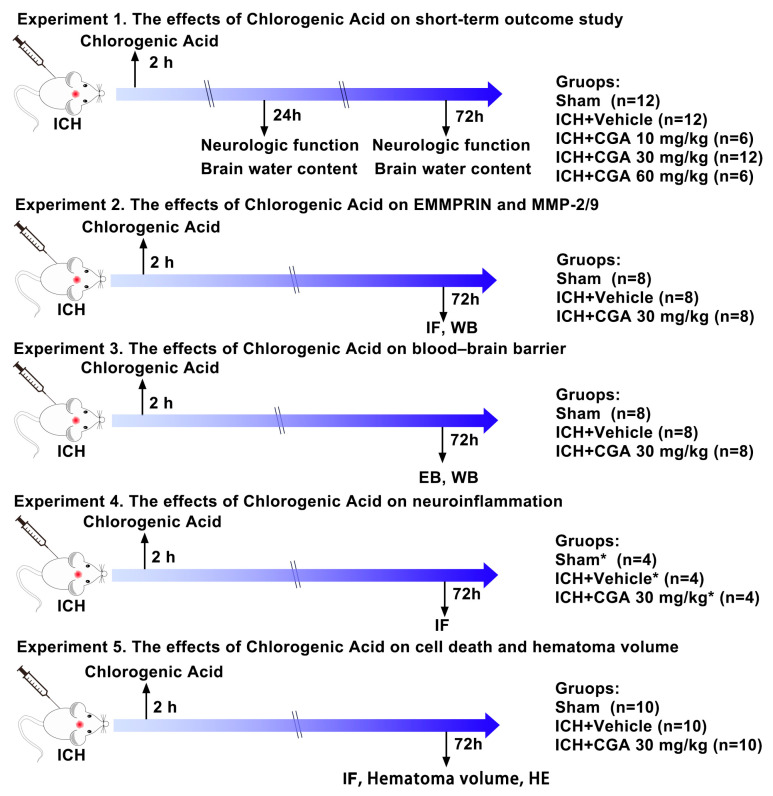
Outline of experimental design. ICH, intracerebral hemorrhage; IF, Immunofluorescence; WB, Western blot; EB, Evans blue; Hematoxylin and Eosin: HE. The asterisk indicates samples shared with experiment 2.

**Figure 2 biomolecules-12-01020-f002:**
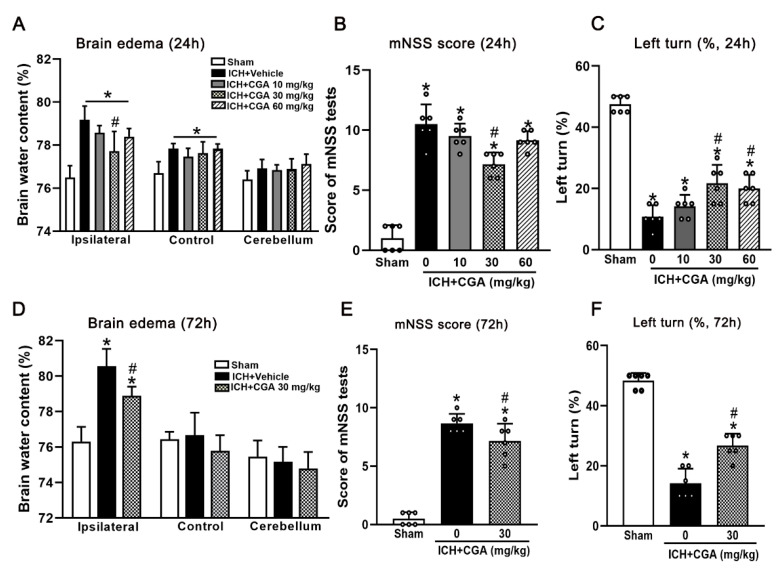
The effects of CGA on brain water content and neurobehavioral outcomes after ICH. (**A**) Brain water content. (**B**) Score of mNSS tests at 24 h. (**C**) Left Turn test at 24 h after ICH. (**D**) Brain water content, (**E**) Score of mNSS tests at 72 h. (**F**) Left Turn test at 72 h after ICH. * *p* < 0.05 vs. sham, # *p* < 0.05 vs. ICH + vehicle. Data are presented as the mean ± SD. *n* = 6 per group.

**Figure 3 biomolecules-12-01020-f003:**
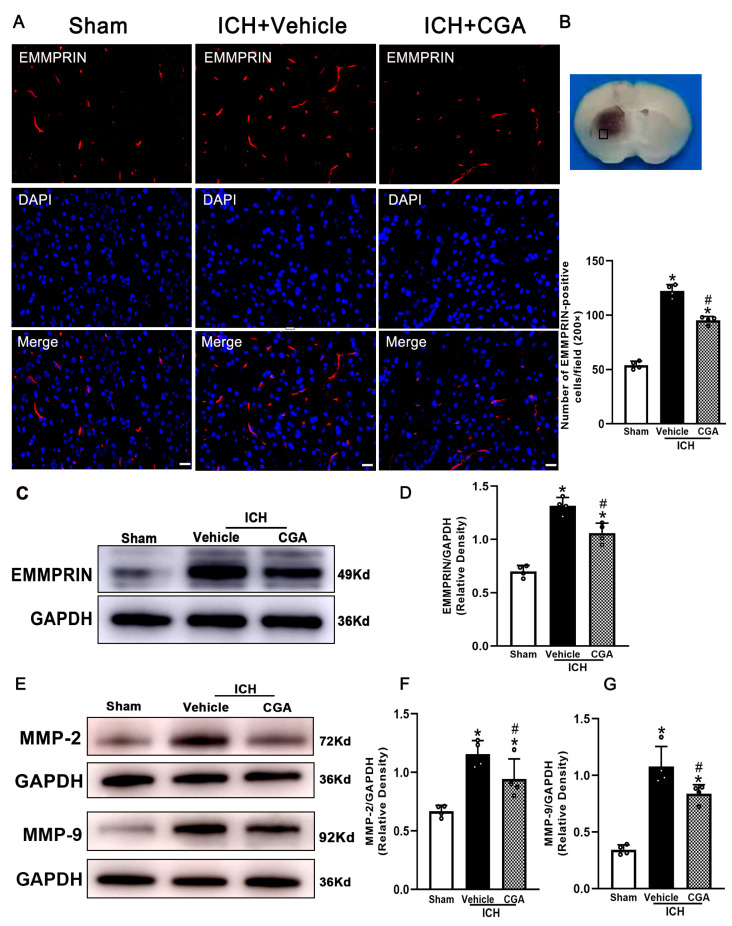
CGA attenuates EMMPRIN and MMP-2/9 expression in mice subjected to ICH. (**A**) Immunostaining of EMMPRIN expression in the perihematomal region at 3 days after ICH. (**B**) The bar graph shows that CGA treatment reduces the number of EMMPRIN cells. (**C**) Western blotting was performed to detect protein levels of EMMPRIN, which showed that CGA reduced the expression of EMMPRIN at 3 days after ICH. (**D**) The bar graph indicates the density of EMMPRIN versus GAPDH in different groups. (**E**) Western blotting was performed to detect protein levels of MMP-2 and MMP-9. CGA reduces the expression of MMP-2 and MMP-9 at 3 days after ICH. (**F**,**G**) The bar graph indicates the density of MMP-2 and MMP-9 versus GAPDH in different groups. * *p* < 0.01 vs. sham; # *p* < 0.05 vs. ICH + vehicle group. Data are presented as mean ± SD; *n* = 4 for each group; scale bar = 20 μm.

**Figure 4 biomolecules-12-01020-f004:**
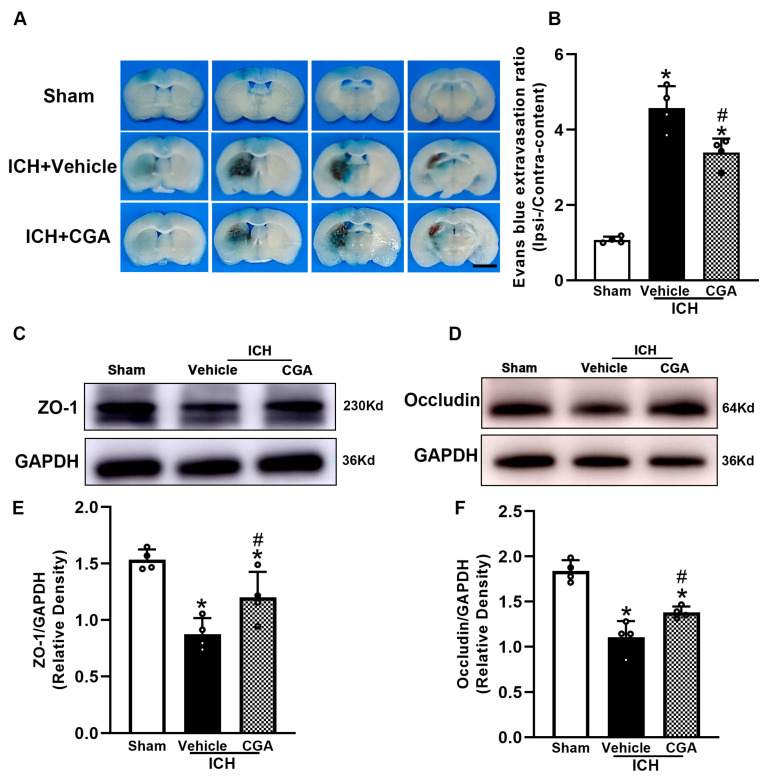
CGA reduces blood–brain barrier permeability in mice subjected to ICH. (**A**) The morphology of brain tissue is shown to observe the effects of CGA on reduction of Evans blue dye leakage at 3 days after ICH, scale bar = 2.5 mm. (**B**) The bar graph indicates that CGA diminishes Evans blue extravasation. (**C**,**D**) Western blotting was performed to detect protein levels of ZO-1 and occludin. CGA reduces the expression of ZO-1 and occludin at 3 days after ICH. (**E**,**F**) The bar graph indicates the density of ZO-1 and occludin versus GAPDH in different groups. * *p* < 0.05 vs. sham; # *p* < 0.05 vs. ICH + vehicle group. Data are presented as mean ± SD; *n* = 4 for each group.

**Figure 5 biomolecules-12-01020-f005:**
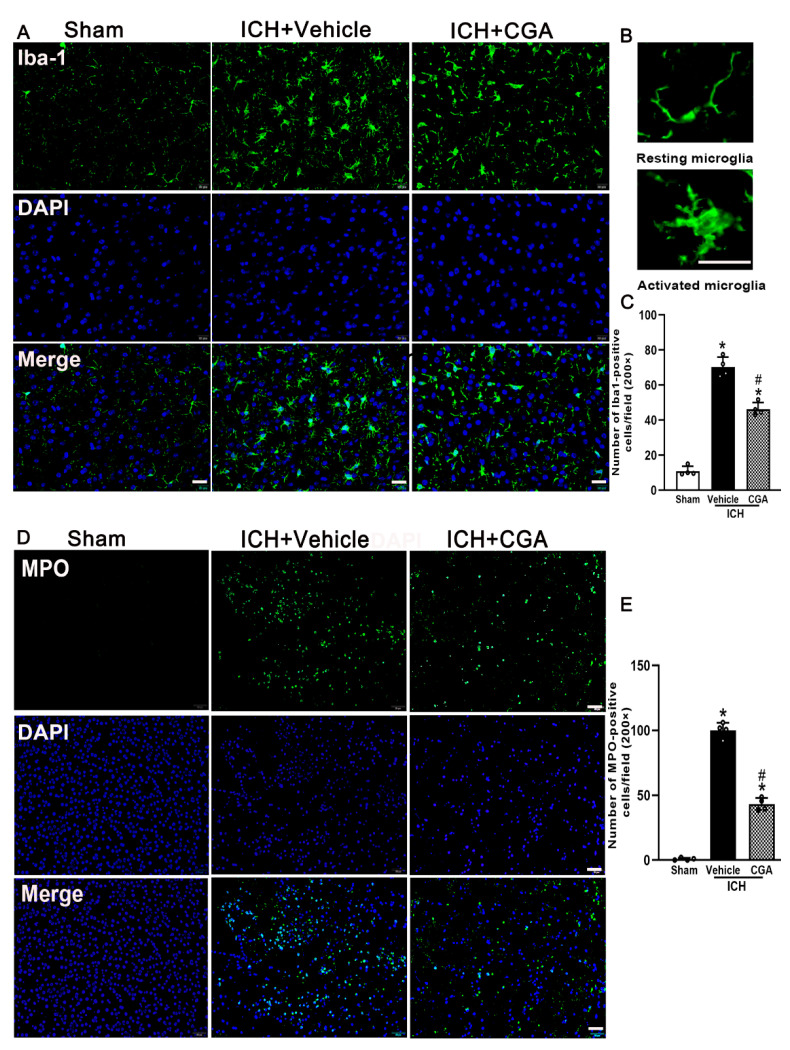
The effects of CGA on microglia/macrophage activation and neutrophil infiltration at 3 days after ICH. (**A**) Representative photographs of immunofluorescent staining for Iba-1-positive microglia. Scale bar = 20 μm. (**B**) Representative photographs of the morphology of resting (up) and activated (down) microglia. Scale bar = 20 μm. (**C**) Quantification analysis shows that CGA significantly decreased the number of activated microglia. (**D**) Representative photographs of immunofluorescent staining for MPO-positive neutrophils. Scale bar = 20 μm. (**E**) Quantification analysis shows that CGA significantly reduced neutrophil infiltration (* *p* < 0.01 vs. sham; # *p* < 0.01 vs. ICH + vehicle group). Data are presented as mean ± SD; *n* = 4 for each group.

**Figure 6 biomolecules-12-01020-f006:**
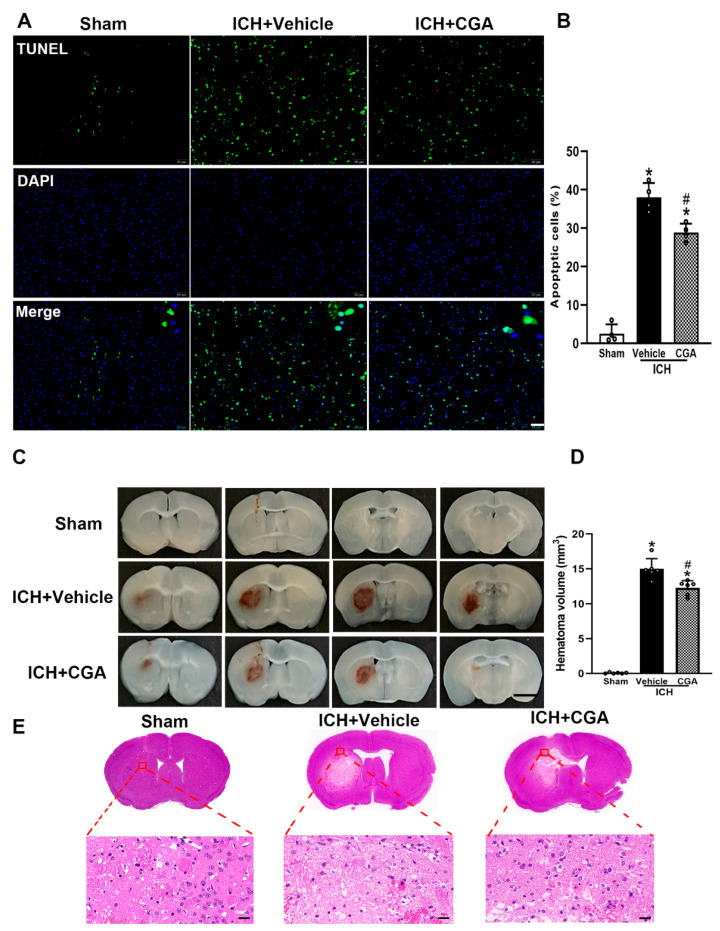
The effects of CGA on cell death and hematoma volume at 3 days after ICH. (**A**) Representative photographs of immunofluorescent staining for TUNEL-positive cells in the perihematomal area. Scale bar = 50 μm, *n* = 4 for each group. (**B**) Quantification analysis shows that CGA significantly decreased numbers of TUNEL-positive cells. (**C**) Representative images of ICH brain for hemorrhagic lesion in response to CGA or vehicle treatment. (**D**) Quantification analysis showed that CGA reduced hematoma volume. Scale bar = 2.5 mm. (**E**) Brain injury was evaluated by HE staining. Scale bar = 20 μm. * *p* < 0.01 vs. sham; # *p* < 0.01 vs. ICH + vehicle group. Data are presented as mean ± SD; *n* = 6 for each group.

## Data Availability

The data in this study could be available from the corresponding author.

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
