# Peer review of "Neuroprotective Effects of Chlorogenic Acid in a Mouse Model of Intracerebral Hemorrhage Associated with Reduced Extracellular Matrix Metalloproteinase Inducer"

_biomolecules, 2022, doi:10.3390/biom12081020_

Round 1

Reviewer 1 Report

The authors present a manuscript on the influence of chlorogenic acid in a mouse model of intracerebral hemorrhage. The authors should point out in the title that the statement in question is based on animal experimental data.

Despite interesting basic idea and good documentation of the experiments, some basic things are missing in this paper. The most important is certainly the power calculation. An animal-experimental approach without prior case number determination is unethical. The authors must be able to show a detailed justification of their experimental animal counts here. 

Another important downside is the fundamental assumption that inflammatory responses are inherently bad in the context of ICH. We now know that much of the repair effort is made possible by the inflammatory milieu in the first place. This fact is completely ignored by the authors. If the "main causes of ICH progression" is supposed to be inflammation (page 12, line 315-16), why do the authors see it in its full expression within only 36 hours? Is the progression not to be observed longer? I also consider the statement about progression of hemorrhage in connection with inflammation to be highly questionable and not practice-related.

Reviewer 2 Report

This study evaluated the effects of  CGA on neuroinflammation and neuronal apoptosis after inhibition of EMMPRIN in a collagenase-induced ICH mouse model.

The topic is of interest; but several points need to be addressed by authors 

1)    There are many grammatical and spelling errors throughout the manuscript. Please, have the paper carefully edited and corrected.

2)    The authors should perform histological analyses to evaluate the brain damage in order to show the centre of haematoma zone with oedema or the peripheral zone with perihaematomal area and oedema, infiltrate of polymorphonuclear cells in peripheral zone, infiltration of macrophage with blood products or red blood cells in peripheral zone 

3)    The authors should report why they choose these doses of CGH

4)    The authors should report the limitations of study

5)    The authors should perform behavioral analyses on memory or spazial learning memory (Morris water maze, NOR, or open field) as reported (for example  in this study: Biomedicines 2022, 10, 1448 doi https://doi.org/10.3390/biomedicines10061448

6)     The authors should investigate the molecular pathway involved such as NF-kB 

7)    The authors should ameliorate the quality of figures

Reviewer 3 Report

Major comments

  1. Line 66-67, the researchers declaimed the innovation of this manuscript to try to uncover the precise effects and mechanisms of CGA neuroprotection in ICH when compared with the previously published paper (ref 29). Yet, ref 29 also tests the CGA treatment effect on MMP-2/9 expression. Although this manuscript test the EMMPRIN expression, microglial cell activation, and neutrophil infiltration, the innovation is not obvious to uncover the precise mechanism of CGA neuroprotection. What are the cell types increasing expression of EMMPRIN and MMP-2/9 after ICH, microglial, astrocyte, endothelial cells or neutrophil? Which cell type does the CGA function mainly work on? Since BBB permeability increased, infiltrated neutrophils could also express EMMPRIN and MMP-2/9, does the increased expression solely because of the infiltrated neutrophils? The researchers need to show the EMMPRIN, MMP-2/9 expression intensity, and colocalization with the different cell types by immunofluorescence, not just by western-blot from hemisphere protein homogenate.

  2. The control the researchers didn’t involve is the CGA effect on the ICH contralateral hemisphere (EMMPRIN, MMP-2/9 expression). According to Figure 3, the EMMPRIN and MMP-2/9 do express at a base level in the sham group, the important question is the expression level in ICH contralateral hemisphere and how will the level change after intraperitoneal CGA treatment.

Minor comments

  1. Fig 2A+D, control? contralateral?
  2. The definition of the ipsilateral and contralateral hemisphere. Is the ipsilateral hemisphere the side induced with ICH?
  3. The figure legends are not clear. For Figure 3B, is the black box shows the location of Figure 3A from the perihematomal region? For Figures 3A and 5C, what does the 200x mean? The Figure 3A sham group EMMPRIN cells are obviously less than 50 cells.
  4. In Figure 5D, the pictures need to be changed to better representative ones. According to the DAPI pictures, The Sham group obviously shows the highest cell density, and ICH+CGA the lowest DAPI cell density. All the three group cell densities should be similar.

Round 2

Reviewer 1 Report

The main points of criticism (including power calculation) have only been dealt with superficially and not constructively.

Reviewer 2 Report

The manuscript is now ok.

Reviewer 3 Report

The researchers have answered all my previous questions.